# Characterization of Bulk Phosphatidylcholine Compositions in Human Plasma Using Side-Chain Resolving Lipidomics

**DOI:** 10.3390/metabo9060109

**Published:** 2019-06-08

**Authors:** Jan D. Quell, Werner Römisch-Margl, Mark Haid, Jan Krumsiek, Thomas Skurk, Anna Halama, Nisha Stephan, Jerzy Adamski, Hans Hauner, Dennis Mook-Kanamori, Robert P. Mohney, Hannelore Daniel, Karsten Suhre, Gabi Kastenmüller

**Affiliations:** 1Institute of Bioinformatics and Systems Biology, Helmholtz Zentrum München—German Research Center for Environmental Health, 85764 Neuherberg, Germany; jan.quell@tum.de (J.D.Q.); werner.roemisch@helmholtz-muenchen.de (W.R.-M.); 2Experimental Bioinformatics, TUM School of Life Sciences Weihenstephan, Technical University of Munich, 85354 Freising-Weihenstephan, Germany; 3Research Unit Molecular Endocrinology and Metabolism, Helmholtz Zentrum München—German Research Center for Environmental Health, 85764 Neuherberg, Germany; mark.haid@helmholtz-muenchen.de (M.H.); adamski@helmholtz-muenchen.de (J.A.); 4Institute for Computational Biomedicine, Englander Institute for Precision Medicine, Department of Physiology and Biophysics, Weill Cornell Medicine, New York City, NY 10021, USA; jak2043@med.cornell.edu; 5Institute of Computational Bioinformatics, Helmholtz Zentrum München—German Research Center for Environmental Health, 85764 Neuherberg, Germany; 6ZIEL Institute for Food and Health, Core Facility Human Studies Technical University of Munich, 85354 Freising-Weihenstephan, Germany; thomas.skurk@tum.de; 7Else Kroener-Frensenius-Center of Nutritional Medicine, Technical University of Munich, 85354 Freising-Weihenstephan, Germany; hans.hauner@tum.de; 8Department of Physiology and Biophysics, Weill Cornell Medicine—Qatar, Education City, P.O. Box 24144, Doha, Qatar; amh2025@qatar-med.cornell.edu (A.H.); nis2034@qatar-med.cornell.edu (N.S.); karsten@suhre.fr (K.S.); 9Department of Biochemistry, Yong Loo Lin School of Medicine, National University of Singapore, Singapore 117596, Singapore; 10Institute for Nutritional Medicine, University Hospital Klinikum rechts der Isar, Technical University of Munich, 80992 Munich, Germany; 11Department of Clinical Epidemiology, Leiden University Medical Center, 2333 Leiden, The Netherlands; d.o.mook@lumc.nl; 12Public Health and Primary Care, Leiden University Medical Center, 2333 Leiden, The Netherlands; 13Metabolon, Inc., Morrisville, NC 27560, USA; rmohney@metabolon.com; 14Chair of Nutrition Physiology, TUM School of Life Sciences Weihenstephan, Technical University of Munich, 85354 Freising-Weihenstephan, Germany; contact@hdaniel.de; 15German Center for Diabetes Research (DZD), 85764 Neuherberg, Germany

**Keywords:** metabolomics, lipidomics, phospholipids, isobaric phosphatidylcholines, lipid species, fatty acid composition, platform comparison, harmonization, imputation

## Abstract

Kit-based assays, such as Absolute*IDQ*^TM^ p150, are widely used in large cohort studies and provide a standardized method to quantify blood concentrations of phosphatidylcholines (PCs). Many disease-relevant associations of PCs were reported using this method. However, their interpretation is hampered by lack of functionally-relevant information on the detailed fatty acid side-chain compositions as only the total number of carbon atoms and double bonds is identified by the kit. To enable more substantiated interpretations, we characterized these PC sums using the side-chain resolving Lipidyzer^TM^ platform, analyzing 223 samples in parallel to the Absolute*IDQ*^TM^. Combining these datasets, we estimated the quantitative composition of PC sums and subsequently tested their replication in an independent cohort. We identified major constituents of 28 PC sums, revealing also various unexpected compositions. As an example, PC 16:0_22:5 accounted for more than 50% of the PC sum with in total 38 carbon atoms and 5 double bonds (PC aa 38:5). For 13 PC sums, we found relatively high abundances of odd-chain fatty acids. In conclusion, our study provides insights in PC compositions in human plasma, facilitating interpretation of existing epidemiological data sets and potentially enabling imputation of PC compositions for future meta-analyses of lipidomics data.

## 1. Introduction

Phosphatidylcholines (PCs) are among the major constituents of eukaryotic cell membranes [1,2] and represent important components of lipoproteins [3]. PCs consist of a polar phosphocholine head group, which is connected via a glycerol backbone to two fatty acid side-chains of varying lengths and degree of saturation. The fatty acids are bound to the *sn1* and *sn2* positions of the glycerol backbone, either via two ester (acyl) bonds, or by one ester (acyl) and one ether (alkyl) bond. The abundance of PCs in human tissues and its relative abundance compared to other phospholipid classes, such as phosphatidylethanolamines (PEs), has been shown to play an important role in regulating energy metabolism [4]. Furthermore, PC composition (i.e., fatty acid side-chain lengths and desaturation levels) influences cell membrane properties, such as their fluidity. As an example, the fluidity of membranes that are made up mainly of PC (16:1/16:1) is higher than that of membranes containing mainly PC (16:0/16:1) or PC (16:0/16:0) [5]. In case of PC (18:0/18:1), the fluidity is maximized when the double bond is located in the Δ9-position [6]. Also for free fatty acids, which can be released from PCs by lipases [7], “minor” configurational molecular differences, such as in the position of the double bond in the fatty acid chain, can have a major functional impact. For instance, a balanced omega-6/omega-3 fatty acid-ratio of 4:1 associates with 70% lower mortality of patients with cardiovascular diseases [8].

Today, the availability of modern high throughput lipidomics approaches allows relative quantification of numerous PCs in thousands of blood samples from epidemiological cohorts. In various large-scale genome-wide and metabolome-wide association studies, many of these PCs have been reported to associate with common genetic variants and various diseases such as Alzheimer’s disease [9,10], type 2 diabetes [11,12,13], metabolic syndrome [14], coronary artery disease [15], and gastric cancer [16].

While the lipid side-chain composition of PCs and the exact configuration of contained fatty acids induce major functional differences, most established lipidomics techniques cannot fully differentiate the specific compounds. Depending on the approach, phospholipids are measured to different degrees of structural resolution, starting at the coarsest level of lipid class/lipid species: for example, based on a precursor ion scan of the mass-to-charge ratio (*m*/*z*) 184 (= *m*/*z* of the phosphocholine head group), PCs and sphingomyelin (SM) species can be selected and further characterized by their total *m*/*z* (e.g., PC (731)), which corresponds to the total numbers of carbon atoms and double bonds in the fatty acid side-chains [17]. This approach does not allow any differentiation of PC species at the bond-type level, i.e., PCs containing an ether bond (e.g., PC O-33:1) cannot be distinguished from isobaric diacyl-PC species (e.g., PC 32:1). Nonetheless, measures at the lipid level are often annotated with the assumption that even-numbered fatty acids are more common than odd-numbered fatty acids. For instance, PC (731) with *m*/*z* 731 covers PC 32:1 and the isobaric PC O-33:1 but is frequently labeled as PC 32:1. In contrast, MS-based lipidomics approaches that, for example, analyze fragmentation spectra can provide measures on the fatty acid level, i.e., they can differentiate between PC O-17:0_16:1 and PC 16:0_16:1. However, usually these fatty acid chain resolving techniques still do not provide any detailed information on the fatty acid position (e.g., PC O-17:0/16:1 versus PC O-16:1/17:0) or on the fatty acid structure (e.g., PC O-17:0/16:1(9Z)) [17,18].

The Absolute*IDQ*^TM^ p150 kits (Biocrates Life Sciences AG, Innsbruck, Austria), which has been proven to provide robust, reproducible measurements [19], have generated large data sets of PCs for epidemiological studies [10,11,15,20,21]. This method (same as the extended kit p180) quantifies PCs on the lipid species level and annotates them under the assumption of even-numbered fatty acid chains (e.g., PC aa C32:1 for PC (731) and PC ae C34:1 for PC (745)) with aa indicating two acyl-bound and ae indicating one acyl- and one alkyl-bound fatty acids). As a consequence, only assumptions can be made regarding which specific PCs (i.e., which combination of fatty acid chain lengths and degrees of desaturation) dominate or significantly contribute to the concentration of the measured PC sum (e.g., PC 16:0_16:1, PC 14:1_18:0, and PC O-17:0_16:1 are all examples for possible constituents of PC aa C32:1). Due to the functional implications of fatty acid structures, mechanistic interpretation of observed genetic and disease associations with bulk PC measures is hampered.

In this study, we seek to reveal the fatty acid compositions of PCs in human plasma measured at the lipid species level to facilitate functional interpretation of existing association results. To this end, we combine quantitative data for PCs from the Absolute*IDQ*^TM^ p150 kit and the fatty acid side-chain resolving Lipidyzer^TM^ platform powered by Metabolon^®^ (Durham, USA). PC concentrations from both methods have been determined in 223 human plasma samples of healthy subjects. To test the transferability of identified PC compositions across different studies, we replicate our results in an independent cohort.

## 2. Results

In a set of 223 human plasma samples from healthy subjects of the HuMet study (see Materials and Methods), we quantified 68 PCs at the lipid species level (e.g., PC aa C32:1) using the Absolute*IDQ*^TM^ p150 kit. Each of these PC measures can be composed of theoretically hundreds of different PCs with specific combinations of fatty acids (i.e., PCs at the fatty acid level e.g., PC 16:0_16:1, PC 14:1_18:0, and PC O-17:0_16:1). In a first step, we systematically assembled all PCs of the fatty acid level that would theoretically fall under each of the 68 PC sums in a list to provide a comprehensive description of their qualitative compositions.

In aliquots of the same 223 plasma samples, concentrations of in total 109 PCs at the fatty acid level (i.e., providing the bond type as well as the lengths and degree of desaturation for the two fatty acid chains without differentiating between *sn-1* and *sn-2* positions) have been quantified using the Lipidyzer^TM^ platform. Comparing the measured PC concentrations (reported in μmol/L) from the two targeted metabolomics methods in a second step, we estimated the quantitative composition of 38 of the 68 PC sums for which at least one isobaric PC has been measured on the Lipidyzer^TM^ platform with at least 25% coverage. Identification of the main PC constituent was replicated in the independent QMDiab cohort (see Materials and Methods) for all 19 PC sums, to which more than one constituent on the fatty acid level was assigned. Finally, we estimated the fraction of PC sums that could not be mapped to any measured PC of the fatty acid level.

### 2.1. Qualitative Composition of Phosphatidylcholine Sums

To map PCs resolved at the fatty acid level to PCs resolved at the lipid species level, we systematically distributed the total numbers of carbon atoms and double bonds to two fatty acid chains for all possible isobaric PCs (Table 1). The isobaric sphingomyelin [^13^C_1_]SM was also added as a possible compound, if it has not been quantified with the kit and, thus, has not been subtracted from the PC sum measure in the isotope correction step of the Absolute*IDQ*^TM^ p150 kit processing (see Materials and Methods). To each of the 68 measured PC sums of the lipid species level, we thereby assigned between 150 and 456 theoretically possible isobaric compounds of the fatty acid level (Appendix A). In the Appendix A, we additionally tagged the PCs of the fatty acid level that were considered to be the most probable constituents of the PC sum (2 to 17 PCs per sum) according to Biocrates [22]. In this table, we also highlighted those PCs that have been quantified on the Lipidyzer^TM^ platform in our data set and that were assigned to one of the listed 68 PC sums (89 out of 109 PCs).

### 2.2. Quantitative Composition of Phosphatidylcholine Sums

For 38 (25 PC aa and 13 PC ae) out of the 68 PC sums quantified with the Absolute*IDQ*^TM^ p150 kit, in total, 69 respective PCs of the fatty acid level (Lipidyzer^TM^) were available at sufficient coverage (25%) in the HuMet Study [23] to estimate quantitative compositions. Each PC sum (lipid species level) consists of one to four measured PCs of the fatty acid level, e.g., PC aa C36:2 = PC 16:0_20:2 + PC 18:0_18:2 + PC 18:1_18:1 + R, PC aa C34:3 = PC 14:0_20:3 + PC 16:0_18:3 + PC 18:2_16:1 + R, or PC aa C32:2 = PC 14:0_18:2 + R (Appendix A). Here, R corresponds to the sum of respective non-measured PCs of the fatty acid level that are specified in Appendix A. In the following, we use the fatty acid level of the shorthand notation of lipids for PCs, e.g., PC *x_1_*:*y_1_*_*x_2_*:*y_2_* according to the resolution of the Lipidyzer^TM^ technique [17]. PCs of the lipid species level are labeled according to the Biocrates-specific notation, e.g., PC aa C*x*:*y* or PC ae C*x*:*y*, corresponding to PC *x*:*y* or PC O-*x*:*y* although the measurement did not allow differentiation of PCs on the bond type level; PC aa or PC ae labels were chosen under the assumption of even chain lengths of fatty acids.

In a first step, we calculated ratios q_ij_ to estimate the characteristic proportion of each measured PC of the fatty acid level *i* based on the concentration of the respective PC sum (lipid species level) *j* in the 223 HuMet samples (Table 2). As an example, three measured PCs of the fatty acid level, namely PC 16:0_20:2, PC 18:0_18:2, PC 18:1_18:1, are isobaric constituents of PC aa C36:2 measured at the lipid species level. The means of ratios PC 16:0_20:2/PC aa C36:2, PC 18:0_18:2/PC aa C36:2, and PC 18:1_18:1/PC aa C36:2 over all subjects and time points (including their 5% and 95% confidence intervals) are 0.037 [0.023, 0.058], 0.967 [0.817, 1.14] and 0.099 [0.060, 0.147], respectively (Figure 1a), and, thus, represent estimates for the proportions of these three species in the PC sum. For 27 out of the 38 investigated PC sums, we identified at least one PC of the fatty acid level with a proportion greater than 20%. For three PC sums, we identified at least two PCs of the fatty acid level with q_ij_ > 0.2. As an example, for PC aa C34:3 the constituents PC 14:0_20:3 (0.048 [0.025, 0.073]), PC 16:0_18:3 (0.482 [0.354, 0.617]), and PC 18:2_16:1 (0.555 [0.381, 0.741]) were measured on the fatty acid level (Figure 1b). In ten cases, we found one major component with a proportion greater than 80%. Interestingly, for one PC sum labeled as the acyl-alkyl-PC (i.e., PC-O compound) PC ae C38:3, we found the diacyl-PC with an odd numbered fatty acid chain PC 17:0_20:3 measured on the fatty acid level as a major component of PC ae C38:3 (0.340 [0.225, 0.477]) (Figure 1c). For 11 PC sums, proportions for all measured constituents summed up to 80-120%. In four cases the sum of proportions was larger than 1.2.

Since PC compositions may vary between individuals and depend on fasting state, we calculated and compared the standard deviation (SD) of the ratios q_ij_ of PC constituents separately for different time points (across subjects) and for the different subjects (across timepoints). For this analysis, we focused on baseline and intervention timepoints before and after lipid-related metabolic challenges, namely 36 h fasting (FASTING), physical activity test (PAT), and after ingestion of a lipid-enriched meal OLTT (see Methods), as we expected to observe the largest differences for them. As a result, we observed SD_challenges_ = 0.074 for the average intra-individual variation, and SD_subjects_ = 0.10 for the average inter-individual variation of q_ij_. No significant trends of the ratios q_ij_ between baseline and intervention timepoints were found for any challenge and any of the main PC constituents. However, the proportions of the main constituents of 13 PC sums were significantly different between subjects (Appendix A). When comparing the distributions of ratios q_ij_ considering all 56 timepoints, distributions were significantly different between subjects for 18 PC sums.

### 2.3. Replication of Quantitative Compositions

To test the transferability of the proportions estimated from the HuMet study (healthy male subjects aged 22–33 y of European ethnicity) to other populations and age ranges, we analyzed data from the two lipidomics methods for participants of the Qatar Metabolomics Study on Diabetes (QMDiab) with 151 healthy controls and 154 patients with diabetes (aged 17–82 y). For 37 out of the 38 PC sums that we investigated, data were available in QMDiab. First, we sought replication of the main constituents (largest q_ij_) identified in HuMet in the 31 healthy male participants of QMDiab below 40 years of age. For all 19 PC sums, for which at least two PC constituents were measured on the Lipidyzer^TM^ platform, the same PCs of the fatty acid level were identified as the main constituents in QMDiab as in HuMet. However, for 23 PC sums, distributions of proportions of main PC constituents showed significant difference between the two cohorts (Table 2, Appendix A). In general, estimated proportions (main and others) tended to be higher in samples of QMDiab, compared to HuMet (Figure 2). Largest differences between proportions in both cohorts were identified for PCs containing C20:4 with the exception of the proportions for PC 15:0_20:4 and PC 17:0_20:4.

As QMDiab also includes samples from female participants and diabetes patients, we were able to compare proportions of PC constituents between these heterogeneous subgroups. Only for one PC sum, we observed a significant difference for the main (single) PC constituent PC 15:0_20:4 for PC ae C36:4.

### 2.4. Estimation of the Contribution of Non-Measured Constituents in Phosphatidylcholine Sums

Out of 150 to 456 theoretically possible PCs of the fatty acid level per PC sum, the platform comparison only allowed mapping of one to four of these PCs to the respective sums to determine proportions (Appendix A). To estimate to which extent each PC sum could be explained by measured PCs of the fatty acid level in our study, we generated linear regression models (estimation of the explained variance; e.g., PC aa C36:2 ~ b ⋅ (PC 16:0_20:2 + PC 18:0_18:2 + PC 18:1_18:1)) and Bland–Altman plots (comparison of absolute concentrations between both analytical methods) for the 38 PCs listed in Table 2.

In median, 58% (mean: 52%, SD: 28%) of the variance of PC sums could be explained by the corresponding sum of measured PCs of the fatty acid level (Table 2 and Appendix A). In consequence, 42% of the variance of PC sums was related to further constituents or other factors such as experimental variation.

Since both methods reported metabolite levels in the same quantitative unit (micromolar), we also used Bland–Altman plots to investigate concordance of the PC sum measured on the lipid species level and the sums of constituents measured on the fatty acid level. Notably, here, concordance can only be expected for those PC sums for which all relevant constituents have been measured (Appendix A). In the plots, we related the differences between concentrations of the PC measure of the lipid species level and the sum of concentrations of mapped PCs of the fatty acid level (e.g., PC aa C36:2 – (PC 16:0_20:2 + PC 18:0_18:2 + PC 18:1_18:1)) to the average of the measure of the lipid species level and the sum of mapped measures on the fatty acid level. In general, small mean differences on the *y*-axis and small variation of these differences indicate concurrent measurements of both approaches. Relating these mean differences to the mean concentrations from both techniques on the *x*-axis of Bland–Altman plots enables identification of trends in discordance of measuring methods that depend on absolute values. In 19 of 25 PC aa compositions, the mean of this difference was within 2 SD of the PC aa measure, in 12 of those compositions (e.g., PC aa C40:4 = PC 18:0_22:4 + PC 20:0_20:4, Figure 3a) mean difference was within 1 SD. In 10 cases, in which mean differences were negative, sum of concentrations of PCs of the fatty acid level were larger than those of PCs of the lipid species level (Figure 3b). In contrast to diacyl-PCs, PC ae sums could only be compared to the sum of isobaric diacyl-PCs containing an odd-numbered fatty acid chain, as phospholipid species with ether bonds have been reported on the Lipidyzer^TM^ platform only for phosphoethanolamines in our data set. Those PCs with odd-numbered fatty acids are considered to be less abundant than the corresponding, unreported, alkyl-acyl-PCs. Supporting this expectation, all 13 PC ae sums showed positive mean deviations larger than 2 SD when compared to the sum of mapped measures on the fatty acid level. For several PCs, we observed systematic trends in the concordance between the compared sums depending on absolute concentration (e.g., PC aa C34:4, PC aa C36:4, PC aa C36:5, PC aa C38:0, Appendix A).

## 3. Discussion

Absolute*IDQ*^TM^ p150 and p180 kits have emerged as established measurement kits for quantification of phospholipids, such as PCs, in blood from thousands of participants in large epidemiological cohorts [11,12,15,16,20,24,25,26,27,28] and in projects using a variety of human and non-human sample matrices [29,30,31,32,33]. As the kits quantify PCs on the level of lipid species, each PC measure is composed of isobaric PCs (within mass resolution) with different combinations of fatty acid chain lengths and degrees of desaturation for the two residues. Depending on chain length and desaturation, biochemical properties such as membrane fluidity [5,6] or lipase activity [7] and, thus, the functions of these constituents, can vary significantly. To facilitate functional interpretation of identified associations between PC sum measures and the outcomes of interest, we here characterized PC compositions (i) by assembling all isobaric phosphocholine lipids on the level of fatty acids (with defined numbers of C atoms and double bonds) that, in theory, could be constituents of the measured PC sum, and (ii) by determining the contribution of constituents to the corresponding sum measure for those constituents that have been measured on the Lipidyzer^TM^ platform in 223 human plasma samples through comparison of measurements from both analytical methods.

In a list provided by Biocrates, 2-16 PCs on the fatty acid level are mentioned as the most abundant variants for each of the 68 PC sums that we investigated in the present study [22]. The selection of listed constituents is mainly based on knowledge from human plasma and serum. Some of the assumptions from human blood, such as a higher abundance of fatty acids with an even number of carbon atoms and specific limitations in the lengths of fatty acids, might not hold in other sample types. Also, in human plasma interpretation of genetic and phenotypic associations with PC sums might be limited if only the short-listed PC constituents were considered as possible constituents of the PC sum. For reference, we therefore collected a comprehensive list of all arithmetically possible PC constituents on the fatty acid level for each of the 68 investigated PC sums quantified with the Absolute*IDQ*^TM^ p150 kit. These lists also include PCs with fatty acids that presumably do not exist in human plasma such as very long-chain fatty acids (e.g., C34) but might be present in plant extracts or other sample types. In this reference, we also included isobaric ^13^C isotopomers of sphingomyelins as in cases where the corresponding isobaric PC is of low abundance, the measured signal might be dominated by the first isotope peak of the not measured, “neighboring” sphingomyelin. While still ignoring the positions of double bonds and stereochemistry, our collection in total lists 150 to 456 constituents per PC sum measured with the Absolute*IDQ*^TM^ p150 kit.

In human plasma, the majority of compounds in our reference list indeed are not expected to be present in considerable amounts. Based on the abundance of cellular free fatty acids, it can be assumed that PCs are mainly composed of the even numbered C16:0, C18:0, C18:1, C18:2, and C20:4 fatty acids as the most common fatty acids in human plasma [34,35,36,37]. For 11 PC sums out of 23 PCs labeled as PC aa C32-C40, at least one PC containing two of those fatty acid chains was quantified on the Lipidyzer^TM^ platform in the HuMet cohort. In 8 of the 11 cases, we could confirm the initial assumption as PCs composed of two highly-abundant fatty acids were found at higher proportions than other combinations measured with Lipidyzer^TM^. In 5 cases, those PCs alone accounted for more than 80% of the PC aa measure. However, for three cases, we also found notable exceptions in our data: (i) for PC aa C36:3, PC 16:0_20:3 was more abundant than PC 18:1_18:2; together they explained more than 80% of the PC sum; (ii) for PC aa C38:5, PC 16:0_22:5 was the major constituent (q_ij_ = 0.55), while PC 18:1_20:4 was only a minor constituent (q_ij_ = 0.098); (iii) for PC aa C38:6, PC 16:0_22:6 was by far the most abundant constituent and explained 77% of the PC sum; PC 18:2_20:4 contributed less to the measured PC sum (q_ij_ = 0.019). Although most proportions for constituents containing C20:4 were significantly larger in the independent QMDiab cohort, we identified the same constituents as major contributors to the corresponding PC sums in all three cases.

As no PCs with ether bonds were measured on the Lipidyzer^TM^ platform in the HuMet cohort, we could not test directly whether the assumption that an alkyl-acyl-PC with even-numbered fatty acid chains were major components for the measures labeled as PC ae. The only constituents for 12 of the 13 PC ae sum measures that have been quantified in our data set were PCs containing an odd-numbered fatty acid chain. On average, proportions of these measured constituents were low (mean q_ij_ = 0.13), suggesting that indeed the not measured isobaric PC-O compounds with two even-numbered fatty acids might account for the major part of the PC sums labeled as PC ae. However, interestingly, we identified 3 PC ae sums containing notable amounts of odd-numbered constituents: PC ae C38:3 with q_ij_ = 0.34 for PC 17:0_20:3, PC ae C36:2 with q_ij_ = 0.23 for PC 17:0_18:2, and PC ae C36:1 with q_ij_ = 0.21 for PC 17:0_18:1.

In 1 of the 13 PC ae sums, namely PC ae C40:1, only PC 18:2_22:6 was measured as possible isobaric constituent (following the rule that PC-O x:y is isobaric with PC x:y+7; see Table 1). Notably, the estimated proportion for PC 18:2_22:6 was 0.57 and, thus, this diacyl-PC contributed considerably to the PC sum labeled as the acyl-alkyl-PC PC ae C40:1 in the Absolute*IDQ*^TM^ p150 kit, posing the question whether this measure should rather be annotated with PC aa C40:8.

Due to availability of QMDiab as a replication cohort and of multi-timepoint data in the HuMet cohort, we were able to investigate variation of PC compositions depending on the individual and his/her fasting status. For various PCs, we observed significant differences in their compositions in terms of the estimated proportion of constituents. In particular, constituents that contained arachidonic acid (C20:4) showed higher proportions in QMDiab. We observed the same trend when analyzing QMDiab subsets (male non-diabetic controls, female non-diabetic controls, and diabetic patients) separately, excluding diabetes as the main reason for this observation. Further factors that could underlie the detected differences in C20:4 abundance between the two cohorts could be general health status (QMDiab diabetics and controls were patients in a dermatological clinic), nutrition, ethnicity, and pre-analytic processing among others. However, despite the observed differences in PC compositions, it has to be noted that the same PCs were identified as main constituents consistently in all subsets that we tested, demonstrating the transferability of our qualitative results to other studies in human plasma.

### 3.1. Limitations

A major limitation in our study is that measures on the fatty acid level are only available for a small number of the possible PC constituents, preventing a comprehensive overview of the actual compositions. For most PC sums with large cumulative proportions for the measured constituents, the sum of measured constituents also largely explained the variance in the data (R^2^). However, in some cases, we estimated large proportions, indicating that measured constituents are main parts of the PC sum, but very low R^2^ (e.g., PC aa C42:6 ~ PC 20:0_22:6 with q_ij_ = 0.7 and R^2^ = 5.9%; PC aa C40:3 ~ PC 20:0_20:3 with q_ij_ = 0.5 and R^2^ = 0.2%). The increased variation of proportions q_ij_ observed between subjects may indicate high inter-individual differences in compositions of the respective PC sums. Another reason for the discordance between proportions and explained variance could also result from overestimation of proportions due to differences in extraction and/or ionization efficiency between the two lipidomics approaches (though both approaches are reporting concentrations (micromolar), the measurements are semi-quantitative). Investigation of such differences in more detail would require measures for all possible (or all relevant) constituents. These differences in extraction and measurement procedures could also explain the estimation of proportions larger than 100% in our data. For four PCs, measured concentrations of constituents on the fatty acid level were much larger than the concentrations of the respective PC sums (lipid species level) with proportions q_ij_ = 1.2 to 1.5.

We used the fatty acid side-chain resolving Lipidyzer^TM^ platform to characterize the molecular constituents of PC sums measured on the lipid species level with an unknown composition of chain lengths and double bonds. While characterizing PC sums by their constituents on the fatty acid level enables interpretations in a more precise biochemical context, PCs of the fatty acid level still denote classes of molecules rather than one compound with a defined chemical structure and function as the *sn-1* and *sn-2* positions are not distinguished, and the positions of double bonds and their stereochemistry cannot be determined. Although we could not investigate these structural properties in our study, it has been shown previously that specific side-chains prefer one of the *sn-1* or *sn-2* positions of PCs in human plasma (in 95% of cases 16:0 is bound to the *sn-1* position, 96% of 18:0 to position *sn-1*, 76% of 18:1 to position *sn-2*, 89% of 18:2 to position *sn-2* and almost 100% of 20:4 to position *sn-2*) [35]. Notably, these previous findings can only be used for interpretation if compositions of the PCs are known at least on the fatty acid level.

Although PC compositions were qualitatively similar across cohorts and timepoints in human plasma samples from healthy and diabetic subjects in the two investigated studies, compositions might be different in other diseases. Also PC compositions identified in this study presumably cannot be readily transferred to other tissues or organisms [38].

### 3.2. Conclusions

Absolute*IDQ*^TM^ p150 kit and Lipidyzer^TM^ platform quantify phosphatidylcholines on different levels of structural resolution. By systematically combining measurements from the two approaches, we here unmasked the composition of phosphatidylcholine sum measures in human plasma from healthy subjects. The identified proportions of the main constituents largely replicated in an independent cohort indicating transferability of our results to other studies. Knowing the composition of these PC sums facilitates the interpretation of associations that have been previously identified from the vast amount of data from the Absolute*IDQ*^TM^ p150 kit in large epidemiological cohorts (or that will be identified in new studies) in a more precise biochemical context. For example, we found PC 14:0_22:6 to be a major component of PC aa C36:6, whose blood levels Draisma et al. [15] reported to associate with a genetic variant in the *GATAD2A* gene. Interestingly, also PC ae C38:0, isobar with PC aa C38:7, showed a genome-wide significant association with the same gene. Further associations not reaching genome-wide significance also highlight PCs with C22:5 and C22:6 fatty acids (including PC aa 38:5 and PC ae C40:1) suggesting a link between the gene and the polyunsaturated C22 fatty acid, which was not apparent from the labeled PC names.

In future studies with a specific focus on lipids, new fatty acid side-chain resolving lipidomics approaches such as Lipidyzer^TM^ can be applied, allowing the direct measurement of PC constituents without the need of the here presented inference of PC compositions. Nonetheless, results from our study also might be useful in upcoming studies in which data from Absolute*IDQ*^TM^ kits and Lipidyzer^TM^ platform have to be combined to maximize sample size and statistical power (e.g., in genome-wide association studies) or to relate baseline measurements performed earlier with Absolute*IDQ*^TM^ p150 kit with new lipidomics data from follow-up visits of the same subjects. In these cases, (re-)measurement on a side-chain resolving lipidomics platform might not be an option, in particular if no sample aliquots are left or if the main focus of an included study was on amino acids and carnitines quantified by the Absolute*IDQ*^TM^ kit rather than on phospholipids. Here, the estimated proportions of PC constituents, which we determined in our study, might enable imputation of PCs of the fatty acid level based on concentrations of PC sums and thus might enable combination of studies with data from different platforms.

## 4. Materials and Methods

### 4.1. Human Plasma Samples

Plasma samples were collected in the Human Metabolome (HuMet) study [23]. The cohort consists of 15 healthy male subjects that have been selected based on following criteria: young age (22–33 y), no medication, and no abnormalities in standard clinical parameters. Participants were submitted to a series of metabolic challenges over four days: extended (36 h) fasting (FASTING) ingesting only 2.7 liters of mineral water, standard liquid mixed meal (SLD) in the form of a fiber-free formula drink supplying one third of their individual recommended daily energy, oral glucose tolerance test (OGTT) with 75 g glucose, physical exercise (PAT) with 30 min on a bicycle ergometer at their personal anaerobic power level, an oral lipid tolerance test (OLTT) with SLD with additional 35 g lipids per square meter of body surface area, and a stress test (STRESS) immersing one hand in ice water for a maximum of three minutes. Blood plasma samples were drawn at 56 different time points before, during, and after challenges. All subjects gave their informed consent for inclusion before they participated in the HuMet study. The study protocol was approved by the ethical committee of the Technische Universität München (#2087/08) and corresponds with the Declaration of Helsinki.

In the present study, only 223 samples of four subjects (subject 5–subject 8) were analyzed, as only for those data was available from both lipidomics methods.

The Qatar Metabolomics Study on Diabetes (QMDiab), in which we sought replication of our results, is a multi-ethnic diabetes case-control study, with participants aged between 17 and 82 years. Metabolite levels of 305 non-fasting blood plasma samples (152 male, 153 female) measured with both methods were available [39]. All subjects gave their informed consent for inclusion before they participated in the QMDiab study. The study was conducted in accordance with the Declaration of Helsinki, and the protocol was approved by the Institutional Review Boards of HMC and Weill Cornell Medical College-Qatar (Research Protocol number 11131/11).

### 4.2. Phosphatidylcholine Quantification on the Lipid Species Level (AbsoluteIDQ^TM^ p150 kit)

For quantifying lipid sums, HuMet and QMDiab plasma samples were analyzed using the Absolute*IDQ*^TM^ p150 kit (Biocrates Life Sciences AG, Innsbruck, Austria). Besides PCs (including lysophosphatidylcholines (lysoPCs)) and SMs, carnitines, amino acids, and hexoses are quantified by this targeted metabolomics approach. The corresponding analytical procedures have been described in detail before [19,23]. In brief, 10 µL of human plasma were pipetted onto filter inserts (containing internal standards) in a 96 well plate. The filters were dried under a nitrogen stream, amino acids were derivatized by addition of a phenylisothiocyanate reagent (5%), and samples were dried again. After extraction with 5 mM ammonium acetate in methanol, the solution was centrifuged through a filter membrane and diluted with running solvent. Metabolites were detected by direct infusion to a 4000 QTRAP system (Sciex, Darmstadt, Germany) equipped with a Shimadzu Prominence LC20AD pump and SIL-20AC auto sampler. 76 PC, 14 lysoPC, and 15 SM species were measured in positive multiple reaction monitoring (MRM) scan mode selective for the common fragment ion of the phosphatidylcholine head group (m/z = 184). The isobaric metabolites (within the mass resolution of the MS) PC *x*:*y* and PC O-*x*+1:*y* cannot be distinguished by this technique. Since odd chain lengths are considered rare for free fatty acids, measured PCs are principally labeled under the assumption of an even number of carbon atoms in the fatty acid chains, i.e., PC aa C*x*:*y* (for PC *x*:*y*) is chosen as label in case of even *x* and PC ae C*x*+1:*y* (corresponding to PC O-*x*+1:*y*) in case of odd *x*. Several internal standards were used for quantification of the phospholipid species. Concentrations were calculated by the Absolute*IDQ*^TM^ kit software and reported in μmol/L. During quality control, metabolites with a coefficient of variation (CV) above 25% and metabolites that showed a significant correlation to the run day and had a CV above 20% (in samples of reference plasma measured along with the HuMet samples) were excluded, preserving 68 and 72 PC, 8 and 10 lysoPC, and 13 and 14 SM species from the HuMet and QMDiab samples for further analyses.

### 4.3. Phosphatidylcholine Quantification on the Fatty Acid Level (Lipidyzer^TM^)

For quantifying lipids resolved at the fatty acid level, plasma samples were analyzed on the Lipidyzer^TM^ platform of AB Sciex Pte. Ltd., Framingham, USA, by Metabolon Inc., Durham, NC, USA. The method allows quantification of over 1100 lipid species from 14 lipid subclasses, including lysoPCs, PCs and SMs [40,41,42,43,44]. Lipids were extracted from samples using dichloromethane and methanol in a modified Bligh-Dyer [45] extraction in the presence of internal standards with the lower, organic phase being used for analysis. The extracts were concentrated under nitrogen and reconstituted in 0.25 mL of dichloromethane:methanol (50:50) containing 10 mM ammonium acetate. The extracts were placed in vials for infusion-MS analyses, performed on a Sciex 5500 QTRAP equipped with the SelexION^TM^ differential ion mobility spectrometry (DMS) cell, where ion separation is based on high-field asymmetric wave-form ion mobility mass spectrometry (FAIMS) [46,47]. This additional device added to the electrospray ionization (ESI) source (applied in positive and negative mode) allows scanning or filtering molecules due to their ion mobility in alternating high and low electric fields before entering the mass spectrometer. With this technology a pre-separation of phospholipid classes (in PCs, lysoPCs, SMs, etc.), independent of their m/z ratio can be achieved. DMS-MS conditions were optimized for phospholipid classes. PCs were detected in negative MRM mode, with characteristic mass fragments for the fatty acid side-chains, thus allowing a resolution of the fatty acid composition. In particular, several PC x:y with an odd-chain fatty acid are reported by the Lipidyzer^TM^ platform. These are identified by the characteristic fragment of the odd-chain fatty acid that cannot be released from an isobaric PC-O x+1:y species with even-numbered side-chains. Individual lipid species were quantified based on the ratio of signal intensity for target compounds to the signal intensity for an assigned internal standard of known concentration. For quantification of PCs, 10 stable isotope labeled compounds with a C16:0 fatty acid side-chain in the *sn-1*-position and side-chains in the range of C16:1–C22:6 in the *sn-2*-position were used as internal standards [41,48].

### 4.4. Qualitative Description of Phosphatidylcholine Sums

For assigning PCs measured by Lipidyzer^TM^ to Absolute*IDQ*^TM^ p150 kit PC sums, we first collected all lipid isobars on the fatty acid precision level that can theoretically contribute to the PC *x*:*y* sum measure (annotated with PC aa C*x*:*y* in the Absolute*IDQ*^TM^ p150 kit). To this end, we systematically distributed the total numbers of carbon atoms *x* and double bonds y to the two fatty acid chains, using the shorthand notation PC *x_1_*:*y_1_*_*x_2_*:*y_2_* with *x = x_1_ + x_2_* and *y = y_1_ + y_2_* to indicate that we do not distinguish between the *sn-1* and *sn-2* positions of the side-chains [17]. Additionally, the isobaric compounds PC *x+1*:*y+7*, PC O-*x+1*:*y* and PC O-*x+2*:*y+7* were considered for PC *x*:*y*. Notably, we also included PCs containing odd-chain fatty acids or fatty acids with very high desaturation to generate a comprehensive list of isobaric PCs. The software coming with the Absolute*IDQ*^TM^ p150 kit, corrects concentrations of PC sums for the concentration of the (within the mass resolution) isobaric sphingomyelins [^13^C_1_]SM *x+4*:*y*, if the corresponding SM has been quantified. If the corresponding SM was not quantified by the kit (i.e., no isotope correction has been performed), we included [^13^C_1_]SM *x+4*:*y* in the list of possible lipid species measured under the PC *x:y* sum.

### 4.5. Quantitative Estimation of Phosphatidylcholine Sum Composition

Both platforms report concentrations of lipid measures in μmol/L. To determine the proportion of a PC at the fatty acid level (measured on the Lipidyzer^TM^ platform) within a PC sum (obtained by the Absolute*IDQ*^TM^ p150 kit), we divided the concentration of the PC *j* (Lipidyzer^TM^) by the concentration of the PC sum PC *i* (Absolute*IDQ*^TM^ p150 kit) that includes this PC *j* according to our qualitative assignment (Appendix A): ratio $q_{\mathrm{ijst}}=\frac{\mathrm{conc}.\left({{\mathrm{PC}}_{j:\;\mathrm{fatty}\mathrm{acid}\mathrm{level}}}_{\mathrm{st}}\right)}{\mathrm{conc}.\left({{\mathrm{PC}}_{i:\;\mathrm{lipid}\mathrm{species}\mathrm{level}}}_{\mathrm{st}}\right)}$. The characteristic composition of each PC sum is estimated by the mean q_ij_ of the respective ratios q_ijst_ over all subjects s and time points t. The variation was specified as 5 and 95% confidence interval, in which the 5% interval contains the lowest 5% and the 95% interval the largest 5% of the ratio values $q_{\mathrm{ijst}}$. Additionally, we determined fractions of all measured isobaric PCs of the fatty acid level, only using measures from the Lipidyzer^TM^ platform: e.g., PC 16:0_20:2/(PC 16:0_20:2 + PC 18:0_18:2 + PC 18:1_18:1).

### 4.6. Replication of Estimated Quantitative Phosphatidylcholine Compositions in an Independent Cohort

Replication of estimated proportions q_ij_ (i.e., quotient of PCs of the fatty acid level (*j*) and the respective PC of the lipid species level (*i*)) was tested in the subset of 31 non-diabetic (control) male subjects aged between 17 and 40 years of the QMDiab cohort (to restrict the comparison to subjects more similar to the participants of the HuMet cohort). After running the cross-platform comparison of the selected QMDiab samples (as described above), we compared the identified main constituent for all PC sums, for which at least two PCs of the fatty acid level have been measured. In addition, we used a Kruskal–Wallis test (function ‘kruskal.test’ of R package ‘stats’ version 3.4.0 [49]) to check if the distributions of proportions q of HuMet- or QMDiab-subjects were significantly different (significance level α = 0.05/37 = 0.0014; with 37 testable PC sums, for which data was available in QMDiab).

### 4.7. Variation of Estimated Phosphatidylcholine Compositions Between Subjects and Challenges

To assess the variation of PC composition within subjects, we determined the average standard deviation (SD) of the ratio q_ij_ observed for samples of the same individual at different timepoints for each PC sum *i* and PC constituent *j*. To this end, we considered “baseline” samples (drawn before the challenges), and “intervention” samples (at the timepoint with largest changes of the total metabolome compared to baseline) for each challenge with relation to lipid metabolism: FASTING: baseline at 8 am after 12 h overnight fasting; intervention at 36 h fasting; PAT: baseline at 4 pm (4 h after last meal); intervention after 30 min exercise; OLTT: baseline at 8 am after 12 h overnight fasting; intervention at noon. In addition, central trends of the ratio $q_{\mathrm{ij}}$ for baseline and intervention timepoints were substantiated by a Wilcoxon signed-rank test (function ‘wilcox.test’ with options ‘two sided’ and ‘paired’ of R package ‘stats’ version 3.4.0 [49]).

To assess the variation of PC composition between subjects, we determined the average SD of the ratio q_ij_ observed for samples at the same timepoint (baseline and intervention timepoints of each lipid-related challenge) in different subjects for each PC sum *i* and PC constituent *j*. We also compared the distributions of q_ij_ from all 56 timepoints of subjects using ANOVA (function ‘aov’ of R package ‘stats’ version 3.4.0 [49]) in case of normal distributions (tested by a Shapiro–Wilk test (function ‘shapiro.test’ of R package ‘stats’ version 3.4.0 [49]) and Kruskal–Wallis (function ‘kruskal.test’ of R package ‘stats’ version 3.4.0 [49]) otherwise. Finally, we compared the distributions of ratios q*_ij_* obtained for male controls, female controls and all diabetics in QMDiab. A Kruskal–Wallis test was used to check if distributions were significantly different between those three groups.

### 4.8. Estimation of Unmapped Part of Phosphatidylcholine Sums

To estimate to what extent the PC bulk measure (in μmol/L) is explained by the sum of all measured PC species (in μmol/L) at the fatty acid level, we applied linear models and Bland–Altman analyses. The formulas for linear models were constructed by ${\mathrm{PC}}_{\mathrm{lipid}\mathrm{species}\mathrm{level}}$ ~ *b* ⋅ ∑(${\mathrm{PC}}_{\mathrm{fatty}\mathrm{acid}\mathrm{level}}$) and evaluated by the ‘lm’ function of the R package ‘stats’, version 3.4.0 [49].

In the Bland–Altman plots, the difference of PC sums (lipid species level) and cumulated respective measured PCs of the fatty acid level $\left(conc.\left({\mathrm{PC}}_{\mathrm{lipid}\mathrm{species}\mathrm{level}}\right)-\sum conc.\left({\mathrm{PC}}_{\mathrm{fatty}\mathrm{acid}\mathrm{level}}\right)\right)$ were opposed to the mean of concentrations $\left(\left(conc.\left({\mathrm{PC}}_{\mathrm{lipid}\mathrm{species}\mathrm{level}}\right)+\sum conc.\left({\mathrm{PC}}_{\mathrm{fatty}\mathrm{acid}\mathrm{level}}\right)\right)/2\right)$ of both platforms.

## Figures and Tables

**Figure 1 metabolites-09-00109-f001:**
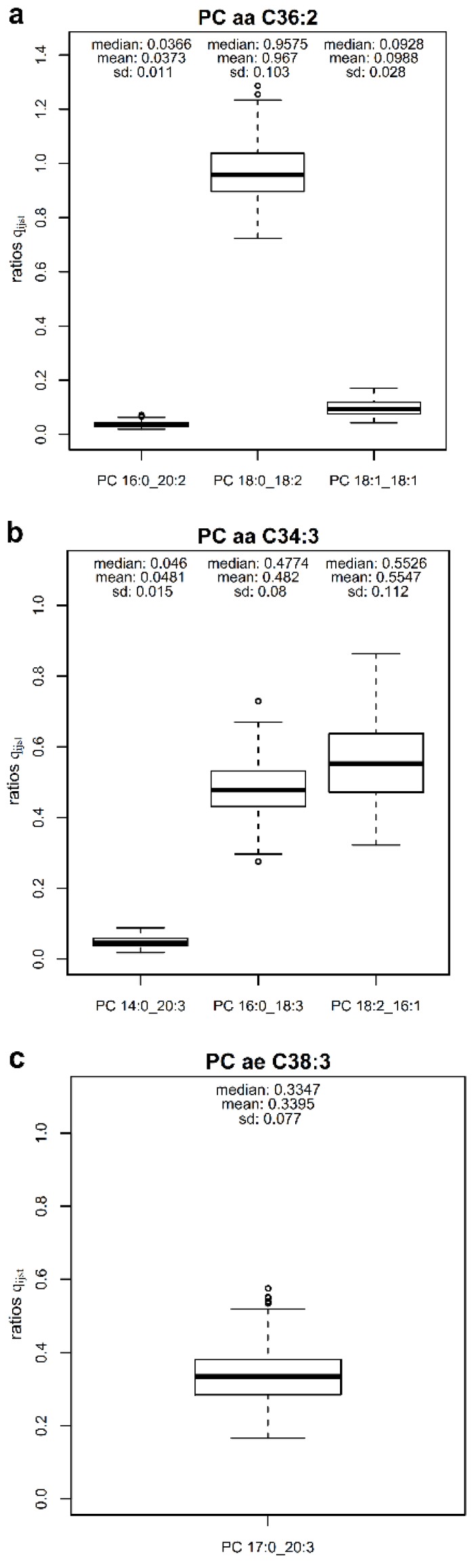
Examples of quantitative compositions of PC sums. The proportion q_ijst_ is defined as the ratio of the measured concentration (μmol/L) of the PC *j* resolved at the fatty acid level (e.g., PC 16:0_18:3) and the concentration (μmol/L) of the respective PC sum *i* at the lipid species level (e.g., PC aa C34:3) in a particular sample. (**a**) On average over all HuMet samples, PC aa C36:2 was mainly composed of PC 18:0_18:2 (96.7%). PC 18:1_18:1 and PC 16:0_20:2 only marginally contributed to the measured concentrations of PC aaC36:2 (9.88% and 3.73%, respectively. (**b**) In contrast to (**a**), PC aa C34:3, mainly consisted of two almost equally abundant PC species, PC 16:0_18:3 (48.2%) and PC 18:2_16:1 (55.5%). (**c**) As concentrations of acyl-alkyl-PCs were not reported for our samples on the Lipidyzer^TM^ platform, the diacyl-PC PC 17:0_20:3 was the only measured constituent of PC ae C38:3, for which we were able to calculate the proportions q_ijst_ in our data set. Interestingly, PC 17:0_20:3 with an odd-chain fatty acid accounted for a considerable fraction (33.95%) of the measured PC ae C38:3 sum.

**Figure 2 metabolites-09-00109-f002:**
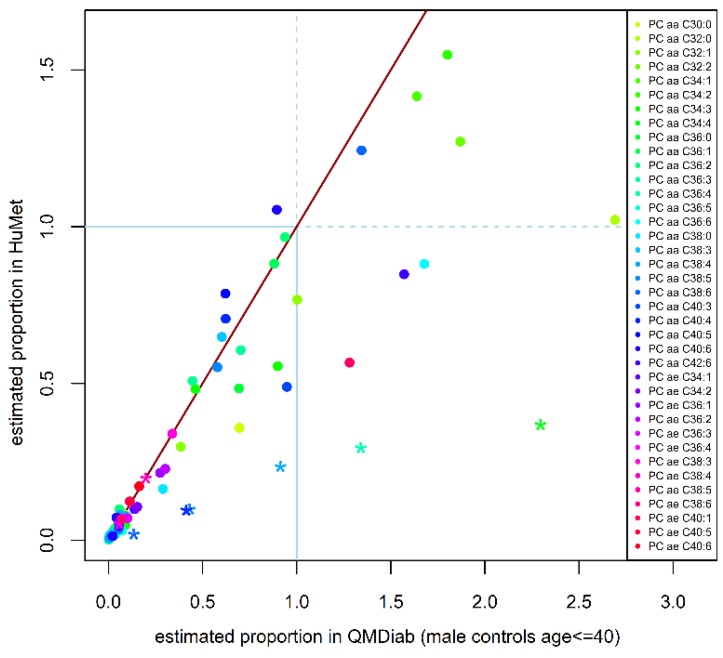
Comparison of characteristic proportions in Human Metabolome (HuMet) study and Qatar Metabolomics Study on Diabetes (QMDiab). On average, proportions tend to be slightly higher in QMDiab than respective proportions in HuMet. The blue lines indicate the threshold where the concentration of the PC constituent measured on the Lipidyzer^TM^ platform was higher than the respective containing PC sum measured with AbsoluteIDQ^TM^ p150 kit. PCs of the fatty acid level that contain C20:4 as one of their lipid side-chains are shown as asterisk.

**Figure 3 metabolites-09-00109-f003:**
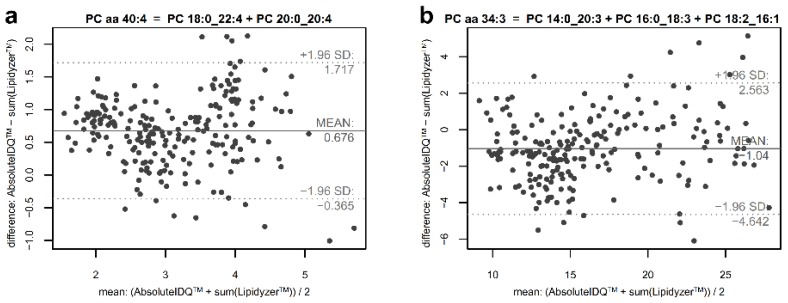
Deviation and mean of PCs of the lipid species level and sum of respective (measured) PCs of the fatty acid level. The Bland–Altman plots show the relative deviation of PC sums (measured on the lipid species level) and the sum of the respective PC constituents (measured on the fatty acid level) dependent on the mean of concentrations. (**a**) In 19 of 25 PC aa measures the mean difference was within 2 SD of the concentration of the PC of the lipid species level. PC aa C40:4 is one example out of 12 of these 19 PC aa, for which the mean difference to its corresponding sum of measured PCs of the fatty acid level (PC 18:0_22:4 + PC 20:0_20:4) was within 1 SD. (**b**) For 10 PC aa, mean differences between the PC sum and the sum of concentrations of PCs of the fatty acid level were negative, i.e., the sum of the components exceeded the concentration measured as PC aa using the Absolute*IDQ*^TM^ kit as shown for the example of PC aa C34:3.

**Table 1 metabolites-09-00109-t001:** Isobaric phosphatidylcholine (PC) sums within mass resolution of the mass spectrometer.

PC Species	Isobaric PC Species	Example	Change in Sum Formula	Change in Mass ^1^
PC x:y		PC 32:0		
	PC x+1:y+7	PC 33:7	+CH_2_ −14H	−0.093900 Da
	PC O-x+1:y	PC O-33:0	+CH_2_ +2H −O	+0.036385 Da
	PC O-x+2:y+7	PC O-34:7	+2(CH_2_) −12H −O	−0.057515 Da
	[^13^C_1_]SM x+4:y	[^13^C_1_]SM 36:0	+^13^C +5H +N −2O	+0.055724 Da
PC O-x:y		PC O-32:0		
	PC O-x+1:y+7	PC O-33:7	+CH_2_ −14H	−0.093900 Da
	PC x-1:y	PC 31:0	+O −CH_2_ −2H	−0.036385 Da
	PC x:y+7	PC 32:7	+O −16H	−0.130285 Da
	[^13^C_1_]SM x+3:y	[^13^C_1_]SM 35:0	+N +H +^13^C −^12^C −O	+0.019339 Da

^1^ Changes in mass have been calculated according to the changes in sum formulas and monoisotopic atom weights w: w(H) = 1.007825 Da, w(C) = 12.000000 Da, w(^13^C) = 13.003355 Da, w(O) = 15.994915 Da, w(N) = 14.003074 Da. In the shorthand notations for lipid structures, x denotes the number of carbon atoms and y the number of double bonds.

**Table 2 metabolites-09-00109-t002:** Quantitative composition of phosphatidylcholine sums. Each PC sum resolved at the lipid species level (Absolute*IDQ*^TM^ p150 kit) consists of a composition of PCs resolved at the fatty acid level (Lipidyzer^TM^). A linear model estimated R^2^ as the percentage of variance of the PC sum that can be explained by the variances of the PCs of the fatty acid level. Proportions correspond to the mean of ratios of the PC at the fatty acid level (measured on the Lipidyzer^TM^ platform) within the matching PC sum (obtained by the Absolute*IDQ*^TM^ p150 kit) over all samples. The percentiles represent (5% and 95%) confidence intervals of proportions. The categories “I” and “II” correspond to proportions larger than 0.2 or larger than 0.8 respectively.

Lipid Species ^1^	AbsoluteIDQ^TM^	Lipidyzer^TM^	R^2^ of	Prop.	Confidence interval		Category	Sum Prop.
(Neutral Mass)	Metabolite	Metabolite	LM [%]	q_ij_	5%	95%		
PC 30:0 (705)	PC aa C30:0	PC 16:0_14:0	83.3	0.3584	0.2248	0.5098	I	0.4404
		PC 18:0_12:0		0.0820	0.0462	0.1495		
PC 32:0 (733)	PC aa C32:0	PC 16:0_16:0	35.3	1.0218	0.7561	1.3897	II	1.0449
		PC 18:0_14:0		0.0231	0.0138	0.0347		
PC 32:1 (731)	PC aa C32:1	PC 14:0_18:1	95.1	0.2981	0.1773	0.5843	I	1.0650
		PC 16:0_16:1		0.7669	0.5433	0.9891	I	
PC 32:2 (729)	PC aa C32:2	PC 14:0_18:2	78.9	1.2713	0.8229	1.7371	II	1.2713
PC 34:1 (759)	PC aa C34:1	PC 16:0_18:1	86.6	1.4159	1.1793	1.6821	II	1.4230
		PC 18:0_16:1		0.0057	0.0028	0.0120		
		PC 20:0_14:1		0.0014	0.0009	0.0022		
PC 34:2 (757)	PC aa C34:2	PC 14:0_20:2	49.4	0.0006	0.0004	0.0010		1.5735
		PC 16:0_18:2		1.5482	1.2968	1.8214	II	
		PC 18:1_16:1		0.0247	0.0155	0.0371		
PC 34:3 (755)	PC aa C34:3	PC 14:0_20:3	85.4	0.0481	0.0252	0.0729		1.0848
		PC 16:0_18:3		0.4820	0.3539	0.6165	I	
		PC 18:2_16:1		0.5547	0.3811	0.7406	I	
PC 34:4 (753)	PC aa C34:4	PC 14:0_20:4	53.6	0.3681	0.2438	0.5650	I	0.3681
PC 36:0 (789)	PC aa C36:0	PC 18:0_18:0	14.3	0.4844	0.3066	0.7470	I	0.4844
PC 36:1 (787)	PC aa C36:1	PC 16:0_20:1	79.0	0.0319	0.0231	0.0447		0.9138
		PC 18:0_18:1		0.8819	0.7171	1.0794	II	
PC 36:2 (785)	PC aa C36:2	PC 16:0_20:2	57.9	0.0373	0.0226	0.0581		1.1031
		PC 18:0_18:2		0.9670	0.8172	1.1362	II	
		PC 18:1_18:1		0.0988	0.0601	0.1467		
PC 36:3 (783)	PC aa C36:3	PC 16:0_20:3	80.4	0.6060	0.3434	0.8409	I	1.1303
		PC 18:0_18:3		0.0175	0.0100	0.0279		
		PC 18:1_18:2		0.5078	0.3177	0.7643	I	
PC 36:4 (781)	PC aa C36:4	PC 14:0_22:4	65.2	0.0016	0.0011	0.0025		0.3842
		PC 16:0_20:4		0.2943	0.2393	0.3555	I	
		PC 18:1_18:3		0.0109	0.0033	0.0220		
		PC 18:2_18:2		0.0774	0.0354	0.1558		
PC 36:5 (779)	PC aa C36:5	PC 14:0_22:5	17.2	0.0172	0.0101	0.0262		0.0482
		PC 18:2_18:3		0.0310	0.0113	0.0615		
PC 36:6 (777)	PC aa C36:6	PC 14:0_22:6	32.3	0.8810	0.4947	1.2925	II	0.8810
PC 38:0 (817)	PC aa C38:0	PC 18:0_20:0	34.5	0.1629	0.1193	0.2153		0.1629
PC 38:3 (811)	PC aa C38:3	PC 18:0_20:3	80.7	0.6485	0.4476	0.8496	I	0.6854
		PC 18:1_20:2		0.0197	0.0126	0.0299		
		PC 18:2_20:1		0.0172	0.0070	0.0436		
PC 38:4 (809)	PC aa C38:4	PC 16:0_22:4	58.8	0.0758	0.0504	0.1059		0.3974
		PC 18:0_20:4		0.2343	0.1971	0.2867	I	
		PC 18:1_20:3		0.0808	0.0490	0.1500		
		PC 18:2_20:2		0.0065	0.0028	0.0126		
PC 38:5 (807)	PC aa C38:5	PC 16:0_22:5	67.7	0.5515	0.4380	0.6725	I	0.6847
		PC 18:1_20:4		0.0983	0.0687	0.1286		
		PC 18:2_20:3		0.0349	0.0201	0.0647		
PC 38:6 (805)	PC aa C38:6	PC 16:0_22:6	77.1	1.2431	1.0168	1.5009	II	1.2621
		PC 18:2_20:4		0.0190	0.0118	0.0279		
PC 40:3 (839)	PC aa C40:3	PC 20:0_20:3	0.2	0.4892	0.1980	0.8453	I	0.4892
PC 40:4 (837)	PC aa C40:4	PC 18:0_22:4	67.2	0.7065	0.4936	0.9503	I	0.8017
		PC 20:0_20:4		0.0952	0.0644	0.1461		
PC 40:5 (835)	PC aa C40:5	PC 18:0_22:5	76.5	0.7863	0.6343	0.9819	I	0.8451
		PC 18:1_22:4		0.0588	0.0373	0.0840		
PC 40:6 (833)	PC aa C40:6	PC 18:0_22:6	82.7	1.0540	0.8588	1.2734	II	1.1384
		PC 18:1_22:5		0.0718	0.0429	0.1039		
		PC 18:2_22:4		0.0126	0.0076	0.0186		
PC 42:6 (861)	PC aa C42:6	PC 20:0_22:6	5.9	0.8482	0.5587	1.2468	II	0.8482
PC 33:1 (745)	PC ae C34:1	PC 15:0_18:1	57.8	0.0988	0.0705	0.1373		0.1426
		PC 17:0_16:1		0.0438	0.0307	0.0601		
PC 33:2 (743)	PC ae C34:2	PC 15:0_18:2	7.1	0.1059	0.0698	0.1541		0.1059
PC 35:1 (773)	PC ae C36:1	PC 17:0_18:1	64.4	0.2147	0.1601	0.2878	I	0.2147
PC 35:2 (771)	PC ae C36:2	PC 17:0_18:2	52.8	0.2275	0.1837	0.2799	I	0.2275
PC 35:3 (769)	PC ae C36:3	PC 15:0_20:3	49.9	0.0700	0.0458	0.0950		0.0700
PC 35:4 (767)	PC ae C36:4	PC 15:0_20:4	26.0	0.0526	0.0339	0.0793		0.0526
PC 37:3 (797)	PC ae C38:3	PC 17:0_20:3	64.5	0.3395	0.2248	0.4767	I	0.3395
PC 37:4 (795)	PC ae C38:4	PC 17:0_20:4	64.1	0.1978	0.1495	0.2547		0.1978
PC 37:5 (793)	PC ae C38:5	PC 17:0_20:5	2.2	0.0263	0.0145	0.0408		0.0263
PC 37:6 (791)	PC ae C38:6	PC 15:0_22:6	12.6	0.0677	0.0433	0.1084		0.0677
PC 39:1 (829)	PC ae C40:1	PC 18:2_22:6	20.3	0.5664	0.3782	0.8648	I	0.5664
PC 39:5 (821)	PC ae C40:5	PC 17:0_22:5	19.7	0.1234	0.0708	0.1735		0.1234
PC 39:6 (819)	PC ae C40:6	PC 17:0_22:6	56.3	0.1720	0.1216	0.2262		0.1720

^1^ Since the measurements by Absolute*IDQ*^TM^ cannot distinguish between acyl and alkyl bond types, we used the isobaric PC followed by the lipid class level mass.

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
