# Peer review of "Characterization of Bulk Phosphatidylcholine Compositions in Human Plasma Using Side-Chain Resolving Lipidomics"

_metabolites, 2019, doi:10.3390/metabo9060109_

Round 1
Reviewer 1 Report
The manuscript is well-written and the study is well-represented with both text and figures/tables. The study describes a unique strategy of combining two data sets from two different platforms to provide additional information regarding the PC species present within a sum composition. Most people likely do not have the funds to purchase a Biocrates kit and definitely not analyses run by Metabolon and certainly not both, so the idea of combining both data sets perhaps has some limitations as to who may actually be able to replicate this approach moving forward, but the idea that the paper takes on is unique and offers a new way to think about assessing assignments to specific PCs in sum compositions, which in my opinion, provides merit for publication. I like the discussion on limitations (so that took care of me adding that to this review); however, it might be worth adding a few sentences as to how this approach will be used moving forward in the community (its lasting effect).
On page 2, line 75, I would put relative or semi- before quantitation
On page 2, line 84, For should not be capitalized
Page 6, line 191, it should be depend
Author Response
Dear Reviewer 1,
Thank you for reviewing our manuscript. Please find enclosed our point-by-point response to your comments.
Kind regards,
Gabi Kastenmüller on behalf of all co-authors

Reviewer 2 Report
Dear authors,
I completed the review of the requested manuscript "Characterization of bulk phosphatidylcholine compositions in human plasma using side-chain resolving lipidomics". The manuscript is important to lipidomics community as it is trying to comprehensively quantify lipids to the fatty acids level using Lipidyzer platform. Additionally, the manuscript have studied the similarity and discrepancies between Biocrates kit and Lipidyzer based lipidomics study. Overall, the study has been done well meeting the objectives. Below are some of my queries that I expect authors would address.
1) As the study has already mentioned and also known to lipidomics community that the biological conclusions between summed PC and PCs with fatty acid information might be affected. Alternatively, the PCs with the fatty acid information can lead to more accurate understanding of biology. Now what would be authors conclusion with this study- MetIDQ P150 0r P180 still important for lipidomics study (I am aware of Biocrates plate have other polar metabolites other than lipids in their kit).
2) The authors have suggested the possibility of imputation of PCs based on summed PCs from Biocrates. I do not think the researchers would be opting a tedious way of quantifying lipidomics using two method (Biocrates and Lipidyzer) and performing imputation. It would be clear to all interested community if the authors can clarify in their conclusion what would be the best way to study lipidomics, and definitely studying lipidomics with all the structural information would be the best. I am aware of conflict but it would be best for the other researchers if the conclusion in the manuscript is more explicit based on what was observed.
3) Many clinical studies have been conducted using Biocrates as authors have pointed out? Are the authors implying that all the published should have their results reviewed if any misinterpretation has occured. Authors themselves have compared the two platforms with the same study. How were the biological interpretation deviated from authors previous studies with Biocrates kit and Lipidyzer? I do not think this question is within the scope of this manuscript but it should be helpful in inferring the findings of the study and would be helpful for other interested scientific community.
4) With the mentioning of "possibility of imputation" , do the authors imply correction to all the previously studies using Biocrates kits or is the suggestion for future studies?
Minor comments:
1) Please look for minor typos. For example line # 107 [...PC(14:1_18::0).
2) Please specify what is x and y in Table 1. I think it is number of carbon and number of double bond. Please specify in the respective location.
3) See typo in the 4th column of Table 1.
4) In line # 327, is the lipid annotation PC(18:2_20:4) correct? I think the double bond should total to 5.
Overall, the study is important for the lipidomics and interested scientific community and can be helpful for understanding the weakness of the commercial kit mentioned herewith. The manuscript will also help researchers in choosing the methods or platforms for lipidomics study. I would also recommend authors to more information in the conclusion section based on their findings.
I would suggest manuscript for acceptance with suggested corrections and addressing of the concerns and queries.
Thank you.
Author Response
Dear Reviewer 2,
Thank you for reviewing our manuscript. Please find enclosed our point-by-point response to your comments.
Kind regards,
Gabi Kastenmüller on behalf of all co-authors
